# Vegan Nutrition for Mothers and Children: Practical Tools for Healthcare Providers

**DOI:** 10.3390/nu11010005

**Published:** 2018-12-20

**Authors:** Luciana Baroni, Silvia Goggi, Roseila Battaglino, Mario Berveglieri, Ilaria Fasan, Denise Filippin, Paul Griffith, Gianluca Rizzo, Carla Tomasini, Maria Alessandra Tosatti, Maurizio Antonio Battino

**Affiliations:** 1Scientific Society for Vegetarian Nutrition, Scientific Committee, Via Verdi 10/9, 30171 Mestre (VE), Italy; luciana.baroni@scienzavegetariana.it (L.B.); mario.berveglieri@scienzavegetariana.it (M.B.); denise.flippin@scienzavegetariana.it (D.F.); paul.griffith@scienzavegetariana.it (P.G.); gianlucarizzo@email.it (G.R.); carla.tomasini@scienzavegetariana.it (C.T.); mariaalessandra.tosatti@scienzavegetariana.it (M.A.T.); 2Department of General Medicine, Humanitas San Pio X, Via Francesco Nava 31, 20159 Milano, Italy; silvia.goggi@sanpiox.humanitas.it; 3Neonatology and Pediatric Unit, San Raffaele Hospital, Via Olgettina 60, 20132 Milano, Italy; battaglino.roseila@hsr.it; 4Division of Inherited Metabolic Diseases, Department of Woman’s and Child’s Health, Padua University Hospital, via Orus 2/B, 35129 Padova, Italy; ilaria.fasan@aopd.veneto.it; 5Nutrition and Food Science Group, Department of Analytical and Food Chemistry, CITACA, CACTI, University of Vigo, Vigo Campus, 36310 Vigo, Spain; 6Department of Clinical Sciences, Università Politecnica delle Marche, Via Ranieri 65, 60100 Ancona, Italy

**Keywords:** vegan diets, vegan pregnancy, vegan breastfeeding, vegan infants, vegan children, plant-based diets

## Abstract

As the number of subjects choosing vegan diets increases, healthcare providers must be prepared to give the best advice to vegan patients during all stages of life. A completely plant-based diet is suitable during pregnancy, lactation, infancy, and childhood, provided that it is well-planned. Balanced vegan diets meet energy requirements on a wide variety of plant foods and pay attention to some nutrients that may be critical, such as protein, fiber, omega-3 fatty acids, iron, zinc, iodine, calcium, vitamin D, and vitamin B12. This paper contains recommendations made by a panel of experts from the Scientific Society for Vegetarian Nutrition (SSNV) after examining the available literature concerning vegan diets during pregnancy, breastfeeding, infancy, and childhood. All healthcare professionals should follow an approach based on the available evidence in regard to the issue of vegan diets, as failing to do so may compromise the nutritional status of vegan patients in these delicate periods of life.

## 1. Introduction

Well-planned vegan diets, when based on a wide variety of plant foods and excluding all animal derivatives, can provide adequate nutrition throughout all stages of life, including pregnancy, lactation, infancy, and childhood [1].

As veganism gains popularity worldwide [2], it does so in Italy, as well. Vegans account for 1% of the total population, and the percentage of people making this dietary choice has been growing in the past years [3]. The exact number of vegan mothers and children following a vegan diet in Italy is not known, but it is likely that the percentage is similar to that of the general population. In the United States, 1% of children aged 8–18 years is estimated to be vegan, as is 3.4% of the total American population [4,5].

The Italian Society for Human Nutrition (SINU) approves vegan diets during pregnancy, lactation, infancy, and childhood [6], and strongly suggests that government institutions and health and nutrition organizations provide more educational resources in order to help Italian plant-based eaters.

In this scenario, a panel of experts from the Scientific Society for Vegetarian Nutrition (SSNV) examined the available literature concerning vegan nutrition in pregnant and breastfeeding women, infants, and children in order to summarize the most relevant recommendations for healthcare providers to best serve their vegan patients in these various delicate stages of life.

In 2016, the SSNV also created the Veg Family Network [7], which groups all experts in the field of vegan nutrition for mothers and children in Italy, so that vegan families can easily find skilled healthcare professionals for advice.

## 2. Well-Planned Vegan Diets: Definition

Although often framed in terms of *lacking*, vegan diets are actually *rich in* a wide variety of foods: grains, legumes (including soy and its derivatives), vegetables, fruits, nuts and seeds, vegetable fats, and herbs and spices [1,6].

Concerns about vegan diets during pregnancy, breastfeeding, infancy, and childhood arose in the past [8,9,10], but this was due to the fact that although being categorized as “vegan”, the investigated subjects were following restrictive diets not respecting all the criteria required to define the diet as being well-planned.

These criteria [6,11] are as follows:Consume large amounts and a wide variety of plant foods, emphasizing the intake of whole or minimally processed foods: a vegan diet can be nutritionally adequate when meeting the calorie requirements from a variety of nutrient-dense foods, mainly unprocessed, belonging to all the plant food groups. The only exception being during late pregnancy, infancy, and early childhood, when fiber must be limited.Limit the amount of vegetable fats, as suggested by the Dietary Reference Intakes (DRIs), in order to not displace more nutrient-dense foods nor limit excess calories. Choose vegetable fats carefully, consuming good sources of omega-3 fatty acids and monounsaturated oils, while avoiding trans fats and tropical oils (coconut, palm, and palm kernel oils) in order to emphasize the efficiency of the omega-3 metabolic pathway. The only exception is during infancy and early childhood, when fats should not be limited but should still be carefully chosen.Consume adequate amounts of calcium and pay attention to vitamin D status: good calcium sources should be obtained by increasing the intakes of calcium-rich foods from plant sources. Conversely, as no kind of diet can provide adequate amounts of vitamin D, the recommendations for vitamin D are the same as for the general population.Consume adequate amounts of vitamin B12: the intake of reliable sources of vitamin B12 is fundamental for a well-planned vegetarian diet, as vitamin B12 status can be compromised, over time, in all vegetarian subjects who do not supplement it.

## 3. Vegan Diets During Pregnancy, Lactation, and Childhood

Several scientific societies have released their position statement on vegetarian diets and are favorable to complete plant-based diets during pregnancy, lactation, infancy, and childhood, as long as they are well-planned (see definition above) [1,4,12,13,14].

Pregnant and breastfeeding vegan women can meet all of their nutrient needs on a vegan diet that includes a variety of plant foods and reliable sources of vitamin B12 and vitamin D [15,16,17].

The average birthweight of infants born to vegan mothers does not differ significantly from that of infants of omnivorous mothers. Macrobiotic vegan women, whose diets can be highly restricted in calories and nutrients, in contrast to well-planned vegan diets, give birth to infants whose weights are significantly lower than expected [18].

Following a plant-rich diet during pregnancy may be protective against the development of preeclampsia, pre-gravid obesity, and minimize the exposure to genotoxic agents. It may also protect from the onset of pediatric diseases, such as pediatric wheezing, diabetes, neural tube defects, orofacial clefts, and some pediatric tumors [19].

Breast milk of vegan women following well-planned vegan diets including a reliable source of vitamin B12 [15,20,21] provides adequate nutrition for their breastfed infants.

The growth of vegan preschoolers, children, and adolescents falls within normal range [22,23], except for those following restrictive macrobiotic diets, whose growth rates are reduced [24].

Children following plant-based diets might have a lower risk of developing obesity [25], are less exposed to veterinary antibiotics found in animal-derived foods [26], and show a more favorable anti-inflammatory adipokine profile [27].

Our review examines the available recommendations regarding nutrients which may be critical in a vegan diet during pregnancy, breastfeeding, infancy, and childhood due to higher requirements and particular physiologic conditions [1,4,12,13].

For the purpose of this paper, we will use the term *infants* as referring to children from birth to 12 months of age, and the term *children* from 1 to 17 years of age.

### 3.1. Protein

Protein requirements can be easily met on a vegan diet that includes a variety of plant foods and meets calorie requirements [1,6,15]. Beans, grains, nuts and seeds, and green leafy vegetables are a good source of protein in vegan diets [28].

Soy and its derivatives, pseudocereals (buckwheat, quinoa, and amaranth), lupins, spinach, and hemp seeds have all the essential amino acids in a proportion similar to animal foods [28], and their consumption should be encouraged.

All essential amino acids can be found in plant proteins [28]. If a variety of plant foods is consumed throughout the day then there is no need for combining different protein sources at each meal [1].

Nevertheless, the presence of antinutritional factors and of fiber is responsible for a lower digestibility of plant proteins (on average about 85%) [29], and when protein needs are particularly high, such as during pregnancy, lactation, infancy, and childhood, some precautions are needed.

#### 3.1.1. Pregnancy and Lactation

Protein intakes should be increased by 10% in vegan pregnant and lactating women, as for all adult vegetarians [6,30].

Additional servings of grains, protein-rich plant foods (legumes, soy milk, soy yoghurt, tofu, tempeh, and meat analogs based on wheat or soy protein) and nuts and seeds should be consumed by vegan women during the second and third trimester of pregnancy and during breastfeeding to meet increased protein requirements [15].

#### 3.1.2. Infancy and Childhood

From 6 to 12 months of age breast or formula milk are good sources of protein in addition to solid foods that are gradually introduced [31], and from 1 to 17 years of age a vegan diet can provide adequate protein, even if the Recommended Dietary Allowance (RDA) increases by 15%, as some authors suggest [12,31]. Since such protein requirements are easily reached and at times exceeded in a vegan diet [32], we suggest heeding to this recommendation.

Infants and young children, whose small stomachs cannot contain great quantities of food and whose total muscle mass is limited together with the efficiency of the amino acid pool, may benefit from consuming different plant sources of protein at each meal, or at least from consuming different plant protein sources at intervals shorter than 6 h [33]. This is very easy for them to do, since they often consume small and frequent meals.

In addition to the above-mentioned protein-rich plant foods, during infancy and early childhood, breast milk and plant-based formula milk provide a good amount of protein [33]. Only commercial infant formulas are recommended for vegan infants, and the use of homemade formulas (based on plant-milks and grains) is strongly discouraged, as it has been associated with nutritional problems in infants [31]. Although the content of isoflavones and aluminum in vegan formulas based on soy protein caused some perplexities in the past in regard to possible negative health effects, the available data suggest that modern soy formulas for infants are a safe option [34].

### 3.2. Fiber

Fiber is abundant in plant-foods and cannot be digested by human enzymes in the digestive tract. Soluble fiber is fermented by gut bacteria, thus producing compounds that may be beneficial to human health [35]. Insoluble fiber increases the bulk of ingested food [36].

By interfering with the absorption of protein and fat and increasing the total volume of food, fiber decreases the calorie density of meals [37]. It also promotes satiety, which occurs earlier after meals containing fiber [38,39].

An excess of fiber, limiting food and calorie intake, may be detrimental during late pregnancy, infancy, and early childhood.

#### 3.2.1. Pregnancy

A regular intake of high fiber foods, as happens in vegan diets, affects the gut microbiota richness of pregnant women positively [40], and so helps fight constipation [41].

Fiber consumption should meet the recommended intake for vegan pregnant women [16,17], unless it causes reduction in food intake and difficulty in meeting the higher energy and nutrient requirements [15], especially during the second and third trimesters, when gastric capacity decreases due to the increased abdominal space required by the fetus.

In this case, fruits and vegetable juices, refined grains, peeled beans, and high-protein, high-energy, fiber-free foods such as soy milk, tofu, and soy yoghurt should be preferred.

#### 3.2.2. Infancy and Childhood

Since the growth rate is very high in the first year of life [42], an excess of fiber may interfere with proper growth by reducing the calorie density of meals, by interfering with the absorption of fats and minerals and leading to early satiety. Vegan infants’ meals up to 12 months of age should be as fiber-deprived as possible (e.g., refined-grains, peeled and mashed beans, or well-cooked beans passed through a sieve). Fiber-free foods such as tofu and soy yoghurt, and strained fruits and vegetables should be preferred [43] (p.339–340). Attention to fiber content must be also paid during the second year of life, as growth velocity is still high [42], but after 12 months whole plant foods also participate in reaching the adequacy of the diet.

### 3.3. Omega-3 Fatty Acids

Well-planned vegan diets should satisfy omega-3 fatty acid requirements during pregnancy, lactation, infancy, and childhood [15,16,17,32].

Good plant sources of omega-3 fatty acids include ground flaxseeds and flaxseed oil, ground chia seeds, and walnuts. One serving of omega-3-rich foods provides approximately 2.5 g of alpha-linolenic acid (ALA) [15,32], from which long chain polyunsaturated fatty acids (PUFAs) eicosapentaenoic/docosahexaenoic (EPA/DHA) are then synthetized [44,45].

To maintain an optimal omega-6/omega-3 ratio and favor the conversion of ALA into PUFAs, seed oils rich in omega-6, trans fats (margarine), and tropical oils (coconut, palm, and palm kernel oils) rich in saturated fats should be avoided or strongly limited [46,47].

Inadequate intakes of energy, proteins, and micronutrients may also impair EPA and DHA synthesis [6].

Olive oil has a low influence on the omega-6/omega-3 ratio and, in addition to flaxseed oil, if used as an omega-3 source, should be the only additional oil to use [45,47].

During the delicate phases of pregnancy, breastfeeding, infancy, and early childhood, when the process of ALA conversion may not keep up with the increased DHA requirements, we suggest insisting upon the Italian DRIs, which recommend a supplementary source of preformed DHA [16].

#### 3.3.1. Pregnancy and Lactation

Diets of pregnant and lactating women should include 2 daily servings of omega-3 rich foods in order to meet requirements [15,16,17].

The conversion rate from ALA to PUFA can be insufficient to meet the slightly increased DHA requirements during pregnancy and lactation [16], for which all pregnant or breastfeeding women, including vegans, should supplement 100–200 mg of DHA daily [15,16,48].

Algal-derived DHA is a viable alternative for vegan women [49].

#### 3.3.2. Infancy and Childhood

Fats should be not limited in infancy and early childhood (they can provide up to 40% of total energy), but rather carefully selected in order to obtain an optimal omega-6/omega-3 ratio [16,44,45,50].

Breast milk of women following a well-balanced vegan diet and formula milk are a good source of omega-3 fatty acids [15,32]. Vegan children from 6 to 12 months should continue to receive breast milk or infant formula on demand and consume 1–2 servings daily of omega-3 rich foods, preferably in the form of flaxseed oil, which does not contain fiber [32].

Vegan children from 1 year of age on should meet their omega-3 requirements by consuming 2 servings of omega-3 rich foods daily [16,17,32].

Choosing at least one serving of flaxseed oil per day instead of the other fiber-containing plant omega-3 sources helps reduce fiber content of the diet when necessary [29,31].

DHA requirements are higher during infancy and early childhood, as DHA participates in retina and neural development [51]. A daily DHA supplement of 100 mg is suggested for all children, including vegans, from 6 months to 3 years of age [16].

Algal-derived DHA is a viable option [49] and may be more acceptable to vegan parents.

### 3.4. Iron

Iron content of vegan diets is higher than in lacto-ovo-vegetarian or omnivorous diets [52].

Iron in plant foods, however, is found in the non-heme form, which can be more variably absorbed than iron in the heme form found in meat, fish, and their derivatives (bioavailability of 1–34% and of 15–35%, respectively) [52,53,54]. Conversely, only the absorption of non-heme iron is subject to homeostatic regulation, which may protect plant-based eaters from iron overload, a risk factor for cardiometabolic diseases [54,55].

Dietary factors and cooking practices may influence non-heme iron absorption [52].

Vitamin C and other organic acids (e.g., citric acid, malic acid), carotene, and retinol increase non-heme iron bioavailability [53,56].

Soaking beans and grains, sour leavening, fermentation, and germination all increase non-heme iron bioavailability by reducing phytates, which are iron sequesters [6,52].

#### 3.4.1. Pregnancy and Lactation

All pregnant women are potentially at risk for iron deficiency (from 7–30% of all pregnancies) [57], as iron requirements nearly double during this period of life [16,17].

Although it has been suggested by some authors that all vegans should increase their recommended iron intake up to 80% [1], a well-balanced vegan diet can easily overcome average iron needs [15].

Iron-rich foods such as whole grains, beans, soy and its derivatives, nuts and seeds, and green leafy vegetables should be consumed daily, in combination with a source of vitamin C (or other organic acids from fruit) or beta carotene [1,52,56]. Cooking practices and food preparation techniques that increase iron absorption should be used whenever possible [1,6,52].

Iron supplementation is required in all pregnant women when hemoglobin levels drop below 110 g/L during the first trimester or below 105 g/L during the second and third trimesters of pregnancy [58,59].

Wheat germ and some herbs, such as dried thyme, have good iron content in small volumes [28] and their regular consumption should be encouraged in pregnant vegan women.

During lactation iron requirements drop dramatically, so the attention to iron intake should return to how it was during the pre-pregnancy period [15,16,17].

#### 3.4.2. Infancy and Childhood

All infants are a population at risk for iron deficiency, thus they should receive complementary solid foods rich in iron [14].

Vegan infants can rely on iron-enriched infant cereals, mashed and peeled beans, soy and its derivatives, and nut and seed butter for reaching an optimal iron intake [14,31].

Wheat germ can be added to soy yoghurt or to other pureed solid foods to increase the iron content of infants’ meals, and a vitamin C source, such as a few drops of lemon juice, can improve iron absorption [28,52,56]. Fiber should be limited, as it may impair iron absorption, and all cooking practices and food preparation techniques that enhance iron absorption should be used when preparing infant food [31].

Vegan children older than 1 year of age should include good iron sources (whole grains, legumes, soy and its derivatives, green leafy vegetables, nuts and seeds) at each meal, along with a source of vitamin C or other organic acids, such as lemon juice or fruit.

In this age group, it is also advised to pay attention to cooking procedures and food preparation techniques that decrease the phytate content of the diet [32].

### 3.5. Zinc

Grains, legumes, soy, and nuts and seeds are good plant sources of zinc [6]. However, zinc absorption may be impaired by the phytate and fiber content of such foods [60,61].

Nutritional yeast is a good source of zinc [28], and its consumption is popular among vegans.

The presence of zinc-rich foods and of vitamin C or other organic acids (i.e., from fruits) in the same meal increases zinc absorption [62].

#### 3.5.1. Pregnancy and Lactation

The consumption of a variety of plant foods rich in zinc should be encouraged throughout the day, along with vitamin C or other organic acids sources (i.e., fruits, a few drops of lemon), as well as the adoption of food preparation techniques that decrease the phytate content of foods (soaking and germination of grains and legumes, fermentation, and sour leavening of bread) [6,15].

Although an interference between zinc and iron absorption has been suggested [63], other data do not support this hypothesis [64], so iron supplements can be prescribed to vegan pregnant women when hemoglobin levels drop without the risk of compromising zinc status.

#### 3.5.2. Infancy and Childhood

From 6 to 12 months of age, breast milk and formula milk are good sources of zinc [6]. Zinc-rich foods should be offered at each meal (legumes, nut and seed butters, soy and its derivatives) [32].

Limiting the fiber content of the diet for children up to 24 months of age by choosing refined products or by manually removing it (peeling beans and straining fruits and vegetables) increases zinc absorption [31,32].

The daily consumption of a wide variety of plant foods can meet zinc recommendations in older children [32], and its absorption can be enhanced by the simultaneous presence of vitamin C and organic acids sources (e.g. some fruits as a dessert, a few drops of lemon in the water) in a meal.

Nutritional yeast can be spread over children’s meals (e.g., pasta, soups) for an additional source of zinc intake.

### 3.6. Iodine

Good sources of iodine, an essential mineral for normal thyroid function, are seafood and, in coastal areas, iodine-containing water and mist from the sea [29] (p. 161).

Many inland populations are at risk of iodine deficiency, regardless of their type of diet, so universal salt iodization is recommended worldwide in order to prevent iodine deficiency [65].

#### 3.6.1. Pregnancy and Lactation

Iodized salt is the safest way to reach iodine requirements in vegan pregnant and lactating women [15]. Iodine content of seaweed, a popular iodine source among vegetarians, is highly variable and excessive iodine intake may impair thyroid function in the fetus and after birth [66,67].

Iodine per gram of iodized salt varies among countries.

In Italy, 1 g of iodized salt contains 30 μg of iodine [68], so 1.3 teaspoons (6.5 g) satisfies the Italian Estimated Average Requirement (EAR) for iodine both in pregnant and lactating vegan women, which is 200 μg per day [15,16].

In the United States, 1 g of iodized salt provides 45 μg of iodine [69], so 1 teaspoon (5 g) during pregnancy and 1.3 teaspoons (6.5 g) during lactation meets the US RDA for iodine in vegan women, which are, respectively, 220 μg and 290 μg per day. Although the World Health Organization (WHO) suggests limiting salt intake to 5 g per day in order to control blood pressure levels [70], vegans are at lower risk for hypertension [71], so a slightly higher intake for this short period of life can be considered harmless in this population. If it is necessary to limit salt intake, an algal-derived supplement can be a viable option.

#### 3.6.2. Infancy and Childhood

Infants and young children are a group at risk of iodine deficiency [14], but complementary foods are only iodine-fortified in some countries [72,73]. In infants and young children not consuming salt, 400 to 900 mL of, respectively, breast or formula milk alone can meet iodine requirements [16,17,74]. If using salt (not before 12 months of age), the daily consumption of 3.3 to 5 g of iodized salt per day in Italian vegan children (providing 100 to 150 μg of iodine) and of 2 to 3.33 g per day in US vegan children (providing 90 to 155 μg of iodine) is suggested in order to meet requirements [32]. Alternatively, an algal-derived iodine supplement can be used.

### 3.7. Calcium

Calcium requirements can be met in a vegan diet by choosing plant foods rich in calcium [6,15].

These include most green leafy vegetables low in oxalates, cruciferous vegetables, sesame seeds, almonds, fortified plant-based milks and plant-based yoghurts, soy, tempeh, calcium-set tofu, and dried figs [15].

Calcium from water has a high bioavailability (23.6% to 47.5%) [75], so tap water (average calcium 100 mg/L) and calcium-rich mineral water (300–350 mg/L) may also help vegans in reaching their daily requirements [15].

Calcium intake is not the only determinant of an optimal bone mass density: low dietary sodium and phosphorus intake, exercise, and an optimal vitamin D and B12 status also positively affect bone mineralization [6,76,77,78].

#### 3.7.1. Pregnancy and Lactation

Calcium requirements are higher during pregnancy and lower during lactation [16,17].

Six daily servings of calcium-rich foods can satisfy calcium requirements in pregnant women, although for calorie requirements above 2400 kcal per day calcium needs are almost automatically satisfied by the variety of plant-based foods consumed [15].

#### 3.7.2. Infancy and Childhood

Vegan infants meet most of their calcium requirements through breast or formula milk [31].

In vegan children, including 3 to 5 serving of calcium-rich foods per day is sufficient to meet requirements [32].

### 3.8. Vitamin D

Vitamin D status depends more on sun exposure and supplementation than on dietary intake [79].

If risk factors for low endogenous vitamin D synthesis are present, such as pigmented skin, low sun exposure, or living at northern latitudes, supplementation should be considered in all subjects, possibly after assessing serum 25-OH vitamin D levels [80].

Both vitamin D2 and vitamin D3 are effective in maintaining optimal vitamin D levels at low-medium doses (600–4000 IU), which are the ones we recommend for maintenance of an optimal vitamin D status in this population [81]. Recommendations for supplementing vitamin D in pregnant and lactating women, infants, and children are summarized in Table 1 [82,83,84].

#### 3.8.1. Pregnancy and Lactation

Vitamin D status should be checked prior to conceiving, as vitamin D insufficiency in mothers may negatively impact on their children’s health [85].

Optimal serum 25-OH vitamin D levels for pregnant women are above 75 nmol/L (30 ng/mL) [86].

Most prenatal vitamins contain insufficient vitamin D in order to prevent vitamin D deficiency in the newborn [87], so daily doses of 1000 to 2000 IU per day are suggested and considered safe in pregnant women [16,17,88]. Levels of 25-OH vitamin D must be checked along with calcium, phosphorus, and parathormone (PTH) after at least 6 months from the beginning of the supplementation (according to Table 1) [85,86].

Supplementing more than 4000 IU per day during pregnancy is not considered safe, so high dose boluses (usually >/= 25,000 IU) of vitamin D must be avoided [89].

#### 3.8.2. Infancy and Childhood

Human milk and formula milk are not sufficient to prevent vitamin D deficiency in infants [87,90]. All infants, including vegan infants, should supplement 400 IU of vitamin D daily throughout their first year of life to prevent rickets and vitamin D deficiency later in life [87].

A regular check of calcium homeostasis (25-OH vitamin D, PTH, calcium, and phosphorus) is necessary until normalization of serum 25-OH vitamin D levels occurs after supplementation (according to Table 1).

### 3.9. Vitamin B12

A sufficient amount of vitamin B12 cannot be found in plant-foods [1,6].

Fermented food and seaweed cannot be considered reliable sources of vitamin B12 [91].

The consumption of B12 fortified foods in vegan diets is sometimes suggested as a means in which to ensure a good daily intake of vitamin B12 [1]. Such products, however, are not always available, and even when they are, they must be consumed three times per day in order to provide adequate amounts of vitamin B12 [6]. Therefore, we suggest that all vegans meet their B12 requirements through supplementation.

Optimal monitoring of B12 status includes dosage of serum homocysteine (HCY), methylmalonic acid (sMMA), and holo-transcobalamin II, along with serum total vitamin B12 [92].

Normal B12 status is defined as holo-transcobalamin II > 45 pmol/L, sMMA < 271 nmol/L, and HCY < 10 μmol/L [92].

Serum total B12 is the most common and widespread method to define B12 status and should be considered optimal above 360 pmol/L, if holo-transcobalamin II is not available, as up to this level there is no increase in the markers of functional B12 deficiency [92]. Daily and weekly doses for maintaining already normal B12 levels, as suggested by the Italian Society of Human Nutrition, are reported in Table 2 [6,93].

#### 3.9.1. Pregnancy and Lactation

Since B12 deficiency can occur during pregnancy regardless of the type of diet, because of store depletion due to higher demands [94], an adequate B12 status should be maintained during vegan pregnancy, and the use of a vitamin B12 supplement represents the most reliable way [20]. Milk from breastfeeding vegan mothers provides adequate vitamin B12 in infants only if vegan mothers are supplementing B12 correctly [20]. Although containing 100% of the RDA for vitamin B12, common pre- and postnatal multivitamins are negatively associated with B12 concentration in breastmilk of vegan women, because only a fraction of the B12 they provide is absorbed [6,20]. Pregnant and lactating vegan mothers should be encouraged to take an individual B12, not multivitamin, supplement and dissolve it under the tongue or chew it slowly in order to increase absorption [6,20].

In case of assessed B12 deficiency, there is no consensus regarding dose, route of administration, or form of the vitamin supplement.

The majority of clinical studies suggest starting with high parenteral doses of B12, after which oral treatment is given [95].

In the United States, the usual treatment depends on injection of 1 mg cyanocobalamin daily for the first week, followed by weekly injections for the next month, and then monthly injections [96].

We suggest the following oral supplementation algorithm, described in Table 3, which depends on the actual serum levels of B12, in order to guarantee a daily amount of absorbed B12 corresponding to 5 times the RDA for B12 [16,17].

B12 supplementation should then proceed, according to Table 2, so as to maintain optimal B12 levels. Serum B12, folic acid, HCY, and Cell Blood Count (CBC) should be checked not before 6–8 months from the beginning of the supplementation.

We suggest that B12 status (serum B12, along with HCY, CBC, and folic acid) should be regularly checked throughout pregnancy also in women with optimal B12 levels in the first trimester of pregnancy, and to adjust supplementation schemes according to the laboratory results.

#### 3.9.2. Infancy and Childhood

Vegan infants should start supplementing B12 with the beginning of complementary feeding, at around 6 months of age since, with the introduction of solid foods, the amount of vitamin B12 provided by breast or formula milk decreases. The amount of B12 to supplement daily varies with age and is shown in Table 2 [6].

In the case of B12 deficiency in infants and children, no protocol regarding supplementation exists so far.

Therefore, we calculated an oral supplementation scheme, shown in Table 3, which depends on the child’s age and on the actual serum B12 levels, in order to guarantee daily amount of absorbed B12 corresponding to 5 times the RDA.

B12 supplementation should then proceed in order to maintain optimal B12 levels. Serum B12, folic acid, HCY, and CBC should be checked not before than 6–8 months from the beginning of the supplementation.

## 4. Menu Planning

The VegPlate is a plate-shaped vegetarian food guide designed to respect Italian and US DRIs during pregnancy, lactation, infancy, and childhood, while using only plant-based foods [15,32].

For each calorie requirement, it suggests the number of servings for each food group (grains, protein-rich foods, nuts and seeds, vegetables, fruits, and fats) to include daily in order to automatically reach a nutritionally adequate vegan diet.

With this method a well-balanced vegan diet can be planned by any healthcare professional within minutes, with no further calculations required.

We have provided three sample menus, obtained with the VegPlate method, in the online Appendix A.

## 5. Conclusions

Vegan diets can meet nutrient requirements and can be an appropriate choice for all life stages, including pregnancy, lactation, infancy, and childhood, provided that they are well-planned. In fact, the problems that occurred in subjects excluding all animal components from their diet were related to the incompleteness of the diet, and thus to nutritional deficiencies. In the past, this was due to the categorization of restrictive diets (i.e., the macrobiotic diet) as vegan [24,97,98]. Today, isolated cases of malnutrition in vegan children have been related almost exclusively to the inappropriateness of the diet offered to the infant or to the lack of B12 supplementation [99,100,101].

A well-planned vegan diet is complete when it follows all the criteria that define it as adequate: (i) consumption of a variety of plant foods throughout the day is encouraged, and no plant-based food group is excluded; (ii) attention is centered on the potentially critical nutrients, namely those that cannot be automatically provided by the variety of the foods consumed. Particularly during pregnancy, breastfeeding, infancy, and childhood, critical nutrients include protein, omega-3 fatty acids, iron, zinc, iodine, and calcium. Vegan pregnant and lactating women and vegan parents must be aware of the dietary sources of such nutrients and of the food preparation techniques and cooking practices that enhance their bioavailability. If sun exposure is insufficient or inefficient, vitamin D supplements are required to maintain an optimal vitamin D status. There are no reliable sources of vitamin B12 in plant foods, as such, a B12 supplementation is mandatory for all vegans.

Due to the rapid increase in popularity of vegan diets, healthcare providers must be aware of the characteristics of a complete vegan diet in order to advise their patients correctly. Vegan diets restricting energy intake, excluding one or more food groups, not paying attention to critical nutrients or to vitamin D status, and not supplementing vitamin B12 cannot be considered well-balanced, and may have dangerous health consequences.

This paper summarizes the recommendations made by the Scientific Society for Vegetarian Nutrition (SSNV) concerning vegan diets during these delicate phases of life. Since there are not enough studies to give evidence-based recommendations, the evidence level of such statements is to be considered as expert opinion.

Not following these recommendations can put these vulnerable subjects at clear risk for nutritional deficiencies.

## Figures and Tables

**Table 1 nutrients-11-00005-t001:** Recommended vitamin D supplements for maintaining normal vitamin D levels or for correcting deficiencies in pregnant and lactating women, infants, and children.

	Maintenance	Deficiency Correction
Pregnancy and lactation	1000–2000 IU/day	2000 IU/day for 5 months or 4000 IU/day for 2.5 months
Children <1 month	400 IU/day	1000 IU/day for 6–8 weeks
Children 1–12 months	400 IU/day	1000–3000 IU/day for 6–8 weeks
Children >12 months	600–1500 IU/day	2000–4000 IU/day for 6–8 weeks

**Table 2 nutrients-11-00005-t002:** Recommended vitamin B12 supplements for maintaining already normal B12 levels in pregnant and lactating women, infants, and children.

	Daily Single Dose	Daily Multiple Dose	Weekly Dose
Pregnant and lactating women	50 μg ^1^	2 μg × 3	1000 μg × 2
Children aged 6 months to 3 years	5 μg	1 μg × 2	-
Children aged 4 to 10 years	25 μg	2 μg × 2	-
Children aged 11 years and above	50 μg	2 μg × 3	1000 μg × 2

^1^ during pregnancy, taking this dose in two separate halves can increase B12 bioavailability.

**Table 3 nutrients-11-00005-t003:** Proposal of an oral supplementation scheme for vitamin B12 deficiency in pregnant and lactating women, infants, and children.

	Serum B12 < 75 pmol/L	Serum B12 between 75 and 150 pmol/L	Serum B12 between 150 and 220 pmol/L	Serum B12 between 220 and 300 pmol/L
Pregnant and lactating women	1000 μg/day for 4 months	1000 μg/day for 3 months	1000 μg/day for 2 months	1000 μg/day for 1 month
Children aged 6 months to 3 years	a daily single dose of 250 μg or 3 daily doses of 10 μg for 4 months	a daily single dose of 250 μg or 3 daily doses of 10 μg for 3 months	a daily single dose of 250 μg or 3 daily doses of 10 μg for 2 months	a daily single dose of 250 μg or 3 daily doses of 10 μg for 1 month
Children aged 4 to 6 years	500 μg 4 times/week for 4 months	500 μg 4 times/week for 3 months	500 μg 4 times/week for 2 months	500 μg 4 times/week for 1 month
Children aged 7 to 10 years	500 μg 6 times/week for 4 months	500 μg 6 times/week for 3 months	500 μg 6 times/week for 2 months	500 μg 6 times/week for 1 month
11 years and above	1000 μg/day for 4 months	1000 μg/day for 3 months	1000 μg/day for 2 months	1000 μg/day for 1 month

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
