# Peer review of "Vegan Nutrition for Mothers and Children: Practical Tools for Healthcare Providers"

_nutrients, 2018, doi:10.3390/nu11010005_

Reviewer 1 Report

This paper contains recommendations made by a panel of experts from  the Scientific Society for Vegetarian Nutrition (SSNV), after examining the available literature concerning vegan diets during pregnancy, breastfeeding, infancy and childhood. However, evidence-based approaches need to be more detail about  vegan diets during pregnancy, breastfeeding, infancy and childhood. 

Author Response

We thank Reviewer 1 for the observation.

The authors examined all the available literature in order to summarize the most important recommendations about vegan diets during pregnancy, lactation, infancy and childhood.

As also the other reviewer suggested, the limited number of studies available on the subject cannot qualify our approach as evidence-based.

We erased the word “evidence-based” in the abstract (line 32) and changed the sentence as follows:

“All health care professional should follow an approach based on the available evidence to the issue of vegan diets, and failing to do so may compromise nutritional status of vegan patients in these delicate periods of life”.

We also changed many parts of the text according to the suggestions of Reviewer 2, and we hope this would ameliorate the paper approach, as you kindly suggested.

Reviewer 2 Report

This is an important paper on the implementation of a vegan diet during pregnancy, lactation, infancy and childhood. As the authors stated, failing to give sound advice may compromise nutritional status in this critical period.

However, the major concern of this paper is, that "evidence-based" advise on vegan diets during the above mentioned life-span is not possible to give, since there is a clear lack of studies on 'modern' vegan diets and health. This is reflected by the reference list of this paper citing only secundary literature. Hence, all recommendations on dietary practice have the evidence-level "expert opinion". This fact should be clearly stated in the paper, for example in the introduction or in the conclusion (which is lacking in the paper now, hence the paper ends a little abrupt).

Lines 62-65: Although I in generally agree that vegan diets can meet nutritional requirements, several case-reports on young children with severe nutrient deficiencies show, that in daily practice there is a remaining risk and not all vegans are informed or willing to follow a well-planned vegan diet. In an article in a scientific journal this should be at least menitoned.

Line 67/68: whole or minimally processed food: How is such food defined? What about the - at least in Germany - growing market of vegetarian or vegan meat and sausage alternatives. What about milk alternatives? And is this feasible in daily practice when both parents have a job?

Line 72: Limit the amount of ... vegetable fats: In my opinion, higher fat intake can be acceptable when fat quality is good? (see for example the recent review of Ludwig et al. Dietary fat: from foe to friend? published in Science 2018; 362: 764-770)

What are tropical oil? Palm and coconut oil?

Line 78: Consume adequate amounts of calcium: This sentence could be extended by all other nutrients (consume adequate amounts of vitamin B2, C, magnesium, biotin). What is the rationale for the selection of calcium and vitamin D?

By the way, vitamin B2 has not been mentioned throughout the text although dairy is the main source.

Macrobiotic diets have been mentioned in the text, although this is not a strict vegan diet (fish has been allowed) and plant foods have been restricted with a focus on grain. In my opinion, the caveats again common vegan diets during childhood resulted in the euqalisation of macrobiotic diets (which resulted in major nutrient deficiencies) and vegan diets.

Line 94: plant-based does not mean vegan. REference 19 also include plant-rich dietary pattern. This reference cannot been cited to prove that vegan diets may be protecitve. For example in this review (see my comment on secondary literature above) cited a study from Congo, that preeclampsia occurred more frequently among women who rarely consumed daily servings of vegetables during pregnancy (33.3%) than among women consuming $3 servings of vegetables per day (3.7%).

REference 22 cited a study on vegan diet in Taiwan. The question emerges, whether Taiwanese vegan food pattern are comparable with Italian/European vegan food pattern.

Line 117/188: this results in how many g protein per day in pregnant and lactating women?

Line 119: what is meant with protein-rich plant foods in this context?

Line 122-128: Please move this sentences in the paragraph above (general remarks on protein).

Line 130: I assume the term formula refers to commercial infant and follwo-on formula according to the requirements of the European commission? This should be clearly stated, as - at least in Germany - sometimes mothers try to prepare vegan milk for there infants from almonds or grain. This should be mentioned and discouraged.

In Germany, such vegan formula based on soy. There are some caveats on isoflavone or aluminium content of such formula and health effects of children. Please see Vandenplas 2014, Brit J Nutrition for review.

Line 149: How much is an "excess of fiber"?

Although I agree that a high fiber diet in infancy may be too bulky, I also see a dissent with the protein requirement, when recommending refined grains and peeled beans, as - to my knowledge - the protein content of these foods is reduced compared to the whole food.

Is soy milk a high protein food?

Line 176: tropical oil: see above

Line 169/170: again, this statement is only proven by secondary literature. To my knowledge, there exist no studies on omega-3-fatty-acid status on a vegan diet. Hence, the authors should think about a more careful phrase: 'vegan diets are supposted to be able to satisfy' or something like this. The same applies to all other comparable statements on the potential of vegan diets on satisfying nutrient requirements.

Line 173/174: There are studies and reviews that question a sufficient conversion rate of ALA to EPA/DHA at least in infancy (10.1007/s00394-015-0982-2 or doi.org/10.1017/S0007114509991851).

What about rapeseed oil in vegan diets?

Line 187: to my knowledge, common supplements of DHA are based on fish oil.

Is there an upper limit of polyunsaturated fatty acids? Results a higher intake of PUFA in a higher need of Vitamin E?

Lines 229/230: 'Wheat germ and some herbs, such as thyme, have a good iron content in a small volume and their regular consumption should be encouraged in pregnant vegan women [28].' Please note, that reference 28 is a food composition database. Hince, the reference should be inserted earlier: "Wheat germ and some herbs, such as thyme, have a good iron content in a small volume [28] and their regular consumption should be encouraged in pregnant vegan women . By the way, what is the iron content in thyme, how many samples have been analysed to set this value? How much is thyme is necessary to fullfill the requiremtents? Is this fresh or dried thyme?  

Iodine: The Joint WHO/FAO Expert Consultation on diet, nutrition and the prevention of chronic disease recommended limiting daily salt intake to at most 5 g per day to control blood pressure levels and reduce hypertension prevalence and related health risks in populations (Joint WHO/FAO Expert Consultation, 2003). Hence, 6.5 g/day cannot be recommended, in particular as 80% of salt comes from commercial food products as bread. In Germany, the additional consumption of bread and other products produced with iodized salt is recommended.

What about infants consuming no or less table salt? Reference 14 states that 'iodine [is] critical in some infants and young children, and that some subgroups in this population may be at
the risk of inadequacy'. 
Breast milk, formula and complementary food does not supply enough iodine in any case (10.1038/ejcn.2009.62). Hence, another iodine source is necessary, either algae with a defined iodien content (e.g. Nori) or iodine supplements (e.g. 50 ug/day) (10.1055/s-0030-1262446)

Calcium: What about fortified food, e.g. fortified milk alternatives (soy milk, almond milk)

Vitamin D: Please provide a table with the recommended Vitamin D supplements. The same applies for vitmain B12 supplementation Lines 386 ff and 408ff

Please aggregate table 1 and 2.

Table 3: do not repeat the word recommendation in the tile line. Aggregate columms with the same statement. As it is only a summary of the text before, it should be questioned whether this table is redundant.

Figures: Please explain the use of the figures (segments indicate amout (%) of food per day?)

Figure 1: If this plate has been published before, the reference should be given in the figure legend. Explain what is meant with protein rich food. What about beverages?

Please aggregate figure 1 and 2

Please give sample menus in (supplementary) tables.

Conclusion is lacking (see above). Please state clearly, that there are not enough studies, to give evidence based recommendations and that those who do not follow these recommendations are at clear risk for nutritional deficiencies (here you can cite your reference Fewtrell et al.: Although theoretically a vegan diet can meet nutrient requirements when mother and infant follow
medical and dietary advice regarding supplementation, the risks of failing to follow advice are severe, including irreversible cognitive
damage from vitamin B12 deficiency, and death).

Author Response

Reviewer 2:

Comments and Suggestions for Authors

1-This is an important paper on the implementation of a vegan diet during pregnancy, lactation, infancy and childhood. As the authors stated, failing to give sound advice may compromise nutritional status in this critical period. 

However, the major concern of this paper is, that "evidence-based" advise on vegan diets during the above mentioned life-span is not possible to give, since there is a clear lack of studies on 'modern' vegan diets and health. This is reflected by the reference list of this paper citing only secundary literature. Hence, all recommendations on dietary practice have the evidence-level "expert opinion". This fact should be clearly stated in the paper, for example in the introduction or in the conclusion (which is lacking in the paper now, hence the paper ends a little abrupt).

We thank Reviewer 2 for the observation.

We erased the word “evidence-based” in the abstract (line 32) and changed the sentence as follows:

“All healthcare professional should follow an approach based on the available evidence to the issue of vegan diets and failing to do so may compromise nutritional status of vegan patients in these delicate periods of life”.

We stated this concept in the conclusions as follows:

“This paper summarizes the recommendations made by the Scientific Society for Vegetarian Nutrition (SSNV), concerning vegan diets during these delicate phases of life. Since there are not enough studies to give evidence-based recommendations, the evidence level of such statements is to be considered as expert opinion.

Not following these recommendations can put these vulnerable subjects at clear risk for nutritional deficiencies.”

2-Lines 62-65: Although I in generally agree that vegan diets can meet nutritional requirements, several case-reports on young children with severe nutrient deficiencies show, that in daily practice there is a remaining risk and not all vegans are informed or willing to follow a well-planned vegan diet. In an article in a scientific journal this should be at least mentioned. 

We thank Reviewer 2 for the observation.

We highlighted this concept in the conclusion as follows:

“Due to the rapid increase in popularity of vegan diets, healthcare providers must be aware of the characteristics of a complete vegan diet in order to advise their patients correctly. Vegan diets restricting energy intake, excluding one or more food groups, not paying attention to critical nutrients or to vitamin D status, and not supplementing vitamin B12 cannot be considered well-balanced, and may have dangerous health consequences.”

3-Line 67/68: whole or minimally processed food: How is such food defined? What about the - at least in Germany - growing market of vegetarian or vegan meat and sausage alternatives. What about milk alternatives? And is this feasible in daily practice when both parents have a job? 

We than Reviewer 2 for the observation.

Plant foods are defined as “whole” when they are consumed (after proper cooking, when required) the same way as they are found in nature: whole-grains, legumes, fruits, vegetables, nuts and seeds.

When plant foods undergo minimal processing (i.e grains are ground into flour and combined with water and yeast to make bread, soybeans are soaked, cooked and strained in order to obtain soymilk, etc.) and no sugar or fats are added they have good (and sometimes ameliorated) nutritional properties, and can be defined as minimally processed.

Meat analogs are either derived from gluten or from textured vegetable protein. The former is the isolated protein of wheat, the latter is what is left after the extraction of soybean oil. Although they can be part of a vegan diet for obvious convenience reasons (and this is why they are included in the group of the protein-rich foods of the VegPlate) their nutrient profile, besides for protein, is mediocre.

On the contrary, whole foods are rich in nutrients, vitamins, minerals and phytochemicals and this is why they should be preferred in a well-balanced vegan diet.

Plant-milks are minimally processed foods which have the advantage of being a good source of calcium, when fortified. Their use is encouraged in the text (Section 3.7 Calcium), and only one serving of plant-milks from the VegPlate fortified with calcium provide the same calcium as two servings of calcium-rich-foods. Their practical use is shown in the sample menus, where is also possible to see how quick a vegan meal for a child can be to prepare.

4-Line 72: Limit the amount of ... vegetable fats: In my opinion, higher fat intake can be acceptable when fat quality is good? (see for example the recent review of Ludwig et al. Dietary fat: from foe to friend? published in Science 2018; 362: 764-770).

We thank Reviewer 2 for the observation.

It is recommended that plant-based diets limit vegetable fats to the amounts suggested by the DRIs in order not to displace more nutrient-dense foods in the diet. We changed the sentence as follows:

“Limit the amount of vegetable fats as suggested by the DRIs, in order to not displace more nutrient-dense foods nor limit excess calories. Choose vegetable fats carefully, consuming good sources of omega-3 fatty acids and monounsaturated oils, while avoiding trans fats and tropical oils (coconut, palm and palm kernel oils) in order to emphasize the efficiency of omega-3 metabolic pathway. The only exception is during infancy and early childhood, when fats shouldn’t be limited, but still carefully chosen.”

5-What are tropical oil? Palm and coconut oil? 

We thank Reviewer 2 for the question.

Yes, tropical oils are coconut, palm and palm kernel oil. We now specified in brackets in the text.

6-Line 78: Consume adequate amounts of calcium: This sentence could be extended by all other nutrients (consume adequate amounts of vitamin B2, C, magnesium, biotin). What is the rationale for the selection of calcium and vitamin D?

We thank Reviewer 2 for the question.

Calcium and vitamin D are typically considered critical nutrients in vegan diets (see references 1 and 6).

7-By the way, vitamin B2 has not been mentioned throughout the text although dairy is the main source. 

We thank Reviewer 2 for the observation.

Vitamin B2 is not considered a critical nutrient in vegan diets (see references 1, 6) as it is provided in amounts when a variety of plant foods are consumed (see 15, 32)

8-Macrobiotic diets have been mentioned in the text, although this is not a strict vegan diet (fish has been allowed) and plant foods have been restricted with a focus on grain. In my opinion, the caveats again common vegan diets during childhood resulted in the euqalisation of macrobiotic diets (which resulted in major nutrient deficiencies) and vegan diets. 

We thank Reviewer 2 for the observation. We totally agree. We modified the sentence as follows:

“Macrobiotic vegan women, whose diet can be highly restricted in calories and nutrients, in contrast to well-planned vegan diets, give birth to infants whose weight is significantly lower than expected [18].”

9-Line 94: plant-based does not mean vegan. REference 19 also include plant-rich dietary pattern. This reference cannot been cited to prove that vegan diets may be protecitve. For example in this review (see my comment on secondary literature above) cited a study from Congo, that preeclampsia occurred more frequently among women who rarely consumed daily servings of vegetables during pregnancy (33.3%) than among women consuming $3 servings of vegetables per day (3.7%). 

We thank Reviewer 2 for the observation.

We changed the word “plant-based” to “plant-rich”.

10-REference 22 cited a study on vegan diet in Taiwan. The question emerges, whether Taiwanese vegan food pattern are comparable with Italian/European vegan food pattern

We thank Reviewer 2 for the observation.

We changed reference 22 with the following one, which refers to European Children:

Sanders, T.A.; Purves, R. An anthropometric and dietary assessment of the nutritional status of vegan preschool children. J Hum Nutr 1981, 35, 349–57.

11-Line 117/188: this results in how many g protein per day in pregnant and lactating women?

The Italian DRIs recommend a daily intake of protein of 46 to 99 g in pregnant and lactating women.

The US DRIs recommend a daily intake of protein of 71 g in pregnant and lactating women, so we can consider this value in the previous range

These values, implemented by 10%, would mean a daily intake of protein of 50.6 to 109 g, amounts which are easily met in a well-balanced vegan diet (see reference 15).

12-Line 119: what is meant with protein-rich plant foods in this context?

We thank Reviewer 2 for the question.

As specified in reference 15, protein-rich plant foods are legumes, soy and its derivatives (including tofu, tempeh, soy yoghurt and soymilk) and meat analogs based on wheat or soy protein.

We modified the sentence as follows:

“Additional servings of grains, protein-rich plant foods (legumes, soy milk, soy yoghurt, tofu, tempeh and meat analogs based on wheat or soy protein) and nuts and seeds should be consumed by vegan women during the 2nd and 3rd trimester of pregnancy and during breastfeeding to meet increased protein requirements [15].”

13-Line 122-128: Please move this sentences in the paragraph above (general remarks on protein). 

We thank Reviewer 2 for the observation.

We moved the sentences as suggested.

14-Line 130: I assume the term formula refers to commercial infant and follwo-on formula according to the requirements of the European commission? This should be clearly stated, as - at least in Germany - sometimes mothers try to prepare vegan milk for there infants from almonds or grain. This should be mentioned and discouraged. 

We thank Reviewer 2 for the observation. We added the following sentence in paragraph 3.1.2.

“Only commercial infant formulas are recommended for vegan infants, and the use of homemade formulas (based on plant-milks and grains) is strongly discouraged as it has been associated with nutritional problems in infants [31].”

15-In Germany, such vegan formula based on soy. There are some caveats on isoflavone or aluminium content of such formula and health effects of children. Please see Vandenplas 2014, Brit J Nutrition for review. 

We thank Reviewer 2 for the observation. We added the following sentence in paragraph 3.1.2.

“Although the content of isoflavones and aluminum in vegan formulas based on soy protein caused some perplexities in the past about possible negative health effects, the available data suggest that modern soy formulas for infants are a safe option [34].”

[34] Vandenplas, Y.; Castrellon, P.G.; Rivas, R.; Gutiérrez, C.J.; Garcia, L.D.; Jimenez, J.E.; Anzo, A.; Hegar, B.; Alarcon, P. Safety of soya-based infant formulas in children. Br J Nutr 2014, 111, 1340-60, doi: 10.1017/S0007114513003942.

16-Line 149: How much is an "excess of fiber"? 

We thank Reviewer 2 for the question.

Although no upper limit is set for the intake of fiber, we intend that any amount of fiber which limits food intake during these periods of high calorie and protein requirements is to be considered excessive.

We modified the sentence as follows:

“An excess of fiber, limiting food and calorie intake, may be detrimental during late pregnancy, infancy and early childhood.”

17-Although I agree that a high fiber diet in infancy may be too bulky, I also see a dissent with the protein requirement, when recommending refined grains and peeled beans, as - to my knowledge - the protein content of these foods is reduced compared to the whole food. 

We thank Reviewer 2 for the observation.

Although there is a minimal difference in protein content (less than 10%) between the whole the refined forms of grains and beans, this does not affect the total daily protein intake.

Breast and formula milk still provide most of the calories and protein under 12 months of ages. Besides, we also encourage the use of fiber-free plant foods such as tofu and soy yoghurt, as shown in the sample menu, as well as the addition of nut and seed butters and oils for an extra dose of calories. In fact, when the calorie requirements are met, proteins are used for plastic purposes instead of energetic ones.

18-Is soy milk a high protein food?

We thank Reviewer 2 for the question.

Yes, it is. We did not mention it on purpose in the section 3.1.2, regarding protein sources for vegan infants, not to create confusion between commercial soy-milk and soy base infant formulas.

19-Line 176: tropical oil: see above

We thank Reviewer 2 for the observation.

We specified that tropical oils are coconut, palm and palm kernel oil.

20-Line 169/170: again, this statement is only proven by secondary literature. To my knowledge, there exist no studies on omega-3-fatty-acid status on a vegan diet. Hence, the authors should think about a more careful phrase: 'vegan diets are supposted to be able to satisfy' or something like this. The same applies to all other comparable statements on the potential of vegan diets on satisfying nutrient requirements. 

We thank Reviewer 2 for the observation.

We modified the sentence as follows:

“Well-planned vegan diets should satisfy omega-3 fatty acid requirements during pregnancy, lactation, infancy and childhood [15–17,32].”

In the text we suggest consuming 2 servings per day of omega-3 rich foods, each of them providing on average 2.5 g of ALA. This recommendation also satisfies the PUFAs requirements, assuming that the ALA from one serving is converted into EPA/DHA with an efficiency of about 10%. To make this conversion more efficient, we suggest keeping the omega-6/omega-3 ratio as low as possible.

Reference: Abedi, E.; Sahari, M.A. Long-chain polyunsaturated fatty acid sources and evaluation of their nutritional and functional properties. Food Sci Nutr 2014, 2, 443–463, doi: 10.1002/fsn3.121.

We also added the following sentence:

“During the delicate phases of pregnancy, breastfeeding, infancy and early childhood, when the process of ALA conversion may not keep up with the increased DHA requirements, we suggest to insist on the Italian DRIs, which recommend a supplementary source of preformed DHA [16].”

21-Line 173/174: There are studies and reviews that question a sufficient conversion rate of ALA to EPA/DHA at least in infancy (10.1007/s00394-015-0982-2 or doi.org/10.1017/S0007114509991851). 

We thank Reviewer 2 for the observation.

Taking into account some studies which question an efficient conversion, the Italian DRIs recommend and intake of preformed DHA in all children under 3 years of age. A well-planned vegan diet for infants should also adhere to this recommendation.

Reference: Società Italiana di Nutrizione Umana. Livelli di Assunzione di Riferimento di Nutrienti ed energia per la popolazione Italiana. Available online: http://www.sinu.it/html/pag/tabelle_larn_2014_rev.asp

22-What about rapeseed oil in vegan diets?

We thank Reviewer 2 for the observation.

Canola oil, although richer in omega-3 fatty acids than other oils, has a lower concentration of ALA (and lower ALA/LA ratio) when compared to flaxseed oil. A greater amount would then be needed to meet omega-3 recommendations, and this would result in not respecting the proportions of macronutrients suggested by the Italian DRIs.

23-Line 187: to my knowledge, common supplements of DHA are based on fish oil.

We thank Reviewer 2 for the observation.

Since the growth of popularity of vegan diets, many algal-derived DHA supplements are now available on the market. Here are some examples: Solgar Vegan DHA, Viridian Vegan EPA & DHA Oil, Dr Rath Omega-3 Vegan, Energy Balance Ovega 3 life DHA.

24-Is there an upper limit of polyunsaturated fatty acids? Results a higher intake of PUFA in a higher need of Vitamin E?

We thank Reviewer 2 for the observation.

The US DRIs suggest an intake of LA ranging between 5 to 10% of total energy, and an intake of ALA ranging from 0.6 to 1.2% of total energy. The Italian DRIs suggest an intake, respectively, of 4 to 8% and of 0.5 to 2%. In a well-planned vegan diet (see references 15 and 32) these limits are respected, so we do not suggest a higher intake of vitamin E.

25-Lines 229/230: 'Wheat germ and some herbs, such as thyme, have a good iron content in a small volume and their regular consumption should be encouraged in pregnant vegan women [28].' Please note, that reference 28 is a food composition database. Hince, the reference should be inserted earlier: "Wheat germ and some herbs, such as thyme, have a good iron content in a small volume [28] and their regular consumption should be encouraged in pregnant vegan women . By the way, what is the iron content in thyme, how many samples have been analysed to set this value? How much is thyme is necessary to fullfill the requiremtents? Is this fresh or dried thyme?  

We thank Reviewer 2 for the observation.

Reference 28, the USDA Food Database, has been moved as suggested.

Moreover, we specified that we refer to dried thyme.

Dried thyme contains 123 mg of iron per 100 g, so 1 g of dried thyme (approximately 1 teaspoon) provides 1 mg of iron. Thyme is not, of course, the only way to meet the increased iron requirements in pregnant women, but a regular consumption of a few teaspoon a day sprinkled over pasta, in salads and soups can help vegan women to reach more easily their iron requirements.

26-Iodine: The Joint WHO/FAO Expert Consultation on diet, nutrition and the prevention of chronic disease recommended limiting daily salt intake to at most 5 g per day to control blood pressure levels and reduce hypertension prevalence and related health risks in populations (Joint WHO/FAO Expert Consultation, 2003). Hence, 6.5 g/day cannot be recommended, in particular as 80% of salt comes from commercial food products as bread. In Germany, the additional consumption of bread and other products produced with iodized salt is recommended. 

We thank Reviewer 2 for the observation.

We added the following sentence at the end of paragraph 3.6.1:

 “In the United States 1 g of iodized salt provides 45 g of iodine [68], so 1 teaspoon (5 g) during pregnancy and 1.3 teaspoons (6.5 g) during lactation meets the US RDA for iodine in vegan women, which are, respectively, 220 mcg and 290 mcg per day.

Although the WHO suggests limiting salt intake to 5 g per day in order to control blood pressure levels [70], vegans are at lower risk for hypertension [71], so a slightly higher intake, for this short period of life, can be considered harmless in this population. If it is necessary to limit salt intake, an algal-derived supplement can be a viable option.”

 And added the following new references 
[70] World Health Organization. Tecnical Report Series. Diet, nutrition and the prevention of chronic diseases. Available online: http://apps.who.int/iris/bitstream/handle/10665/42665/WHO_TRS_916.pdf;jsessionid=C58AEB3899AEAE3D6230A99363CE404C?sequence=1 (accessed on 4 december 2018). 

[71] Yokoyama, Y.; Nishimura, K.; Barnard, N.D.; Takegami, M.; Watanabe, M.; Sekikawa, A.; Okamura, T.; Miyamoto, Y. Vegetarian diets and blood pressure: a meta-analysis. JAMA Intern Med 2014, 174, 57787, doi: 10.1001/jamainternmed.2013.14547.

 27-What about infants consuming no or less table salt? Reference 14 states that 'iodine [is] critical in some infants and young children, and that some subgroups in this population may be at
the risk of inadequacy'. Breast milk, formula and complementary food does not supply enough iodine in any case (10.1038/ejcn.2009.62). Hence, another iodine source is necessary, either algae with a defined iodine content (e.g. Nori) or iodine supplements (e.g. 50 ug/day) (10.1055/s-0030-1262446)

We thank Reviewer 2 for the observation. We rephrased the paragraph as follows:

“Infants and young children are a group at risk of iodine deficiency [14], but only in some countries complementary foods are iodine-fortified [72,73]. In infants and young children not consuming salt, 400 to 900 ml of, respectively, breast or formula milk alone can meet iodine requirements [16,17,74]. If using salt (not before 12 months of age), the daily consumption of 3.3 to 5 mcg of iodized salt per day in Italian vegan children (providing 100 to 150 mcg of iodine) and of 2 to 3.33 g per day in US vegan children (providing 90 to 155 mcg of iodine) is suggested in order to meet requirements [32]. Alternatively, an algal-derived iodine supplement can be used.”

And added the following new references

 [72]   Alexy, U.; Drossard, C.; Kersting, M.; Remer, T. Iodine intake in the youngest: impact of commercial complementary food. Eur J Clin Nutr 2009, 63, 1368–70, doi: 10.1038/ejcn.2009.62. 
[73] Remer, T. Johner, S.A.; Gärtner, R.; Thamm, M.; Kriener, E. Iodine deficiency in infancy - a risk for cognitive development. Dtsch Med Wochenschr 2010, 135, 1551–6, doi: 10.1055/s-0030-1262446. 

[74] Dumrongwongsiri, O.; Chatvutinun, S.; Phoonlabdacha, P.; Sangcakul, A.; Chailurkit, L.O.; Siripinyanond, A.; Suthutvoravut, U.; Chongviriyaphan, N. High Urinary Iodine Concentration Among Breastfed Infants and the Factors Associated with Iodine Content in Breast Milk. Biol Trace Elem Res 2018, 186, 106113, doi: 10.1007/s12011-018-1303-4.

28-Calcium: What about fortified food, e.g. fortified milk alternatives (soy milk, almond milk)

We thank Reviewer 2 for the observation.

We mentioned fortified plant-milks in paragraph 3.7 as good calcium sources.

29-Vitamin D: Please provide a table with the recommended Vitamin D supplements. The same applies for vitamin B12 supplementation Lines 386 ff and 408ff

We provided the new tables, as suggested by Reviewer 2, respectively the new tables 1, 2 and 3, and we adjusted the corresponding text.

30-Please aggregate table 1 and 2. 

We thank Reviewer 2 for the suggestion.

They became the new Table 2, and we adjusted the corresponding text.

31-Table 3: do not repeat the word recommendation in the tile line. Aggregate columns with the same statement. As it is only a summary of the text before, it should be questioned whether this table is redundant. 

We thank Reviewer 2 for the observation.

We erased the old Table 3 in order not to be redundant.

32-Figures: Please explain the use of the figures (segments indicate amount (%) of food per day?)

We thank Reviewer 2 for the question.

Reference 15, which we added to the figure, explains the graphic of the VegPlate in detail.

We added the following sentence to the Supplementary Material

“The six slices of the main plate are not proportional to the amount, nor to the percentage of daily calories to consume rom each food group, as the latter changes change according to the different calorie requirements. On the contrary, the slices of the three small plates in Figure 2b are proportional to the relative amount of food to add to the main plate for each group during these stages of life. The graphic of the main VegPlate (Figure 1a) simply illustrates the variety of plant-foods to consume daily in a well-planned plant-based diet, and highlights the importance of including good sources of calcium and omega-3 fatty acids and of supplementing vitamin B12 and vitamin D.”

33-Figure 1: If this plate has been published before, the reference should be given in the figure legend. Explain what is meant with protein rich food. What about beverages?

We thank Reviewer 2 for the question.

In reference 15, which we added to the figure legend, are listed all the protein-rich foods in a vegan diet.

Protein-rich foods include both solid plant foods (legumes, tofu, tempeh, soy yoghurt, meat analogs) and liquid plant foods (soymilk). Plant-milks derived from cereal belong to the cereal group.

34-Please aggregate figure 1 and 2

We aggregated the figures, as suggested by Reviewer 2.

35-Please give sample menus in (supplementary) tables. 

We thank Reviewer 2 for the observation.

We gave the sample menus in a supplementary Table, as suggested by Reviewer 2.

We added the following paragraph in the text:

“4. Menu Planning

The VegPlate is a plate-shaped vegetarian food guide designed to respect Italian and US DRIs during pregnancy, lactation, infancy and childhood, while using only plant-foods [15,32].

For each calorie requirement, it suggests the number of servings for each food group (grains, protein-rich foods, nuts and seeds, vegetables, fruits and fats) to include daily, in order to automatically reach a nutritionally adequate, vegan diet.

With this method a well-balanced, vegan diet can be planned by any healthcare professional within minutes, with no further calculations required.

We provide three sample menus, obtained with the VegPlate method, in the online Supplementary Material.”

36-Conclusion is lacking (see above). Please state clearly, that there are not enough studies, to give evidence based recommendations and that those who do not follow these recommendations are at clear risk for nutritional deficiencies (here you can cite your reference Fewtrell et al.: Although theoretically a vegan diet can meet nutrient requirements when mother and infant follow
medical and dietary advice regarding supplementation, the risks of failing to follow advice are severe, including irreversible cognitive damage from vitamin B12 deficiency, and death). 

We thank Reviewer 2 for the observation.

We modified and implemented the conclusion as follows.

Conclusion

Vegan diets can meet nutrient requirements and can be an appropriate choice for all life stages, including pregnancy, lactation, infancy and childhood, provided that they are well-planned. In fact, the problems that occurred in subjects excluding all animal components from their diet were related to the incompleteness of the diet, and thus to nutritional deficiencies. In the past, this was due to the categorization of restrictive diets, i.e. the macrobiotic diet, as vegan [97-99]. Today, isolated cases of malnutrition in vegan children have been related almost exclusively to the inappropriateness of the diet offered to the infant or to the lack of B12 supplementation [100-102].

A well-planned vegan diet is complete when it follows all the criteria which define it as adequate: i) the consumption of a variety of plant foods throughout the day is encouraged, and no plant-food group is excluded; ii)attention is centered on the potentially critical nutrients, namely those that cannot be automatically provided by the variety of the foods consumed. Particularly during pregnancy, breastfeeding, infancy and childhood critical nutrients include protein, omega-3 fatty acids, iron, zinc, iodine and calcium. Vegan pregnant and lactating women and vegan parents must be aware of the dietary sources of such nutrients and of the food preparation techniques and cooking practices which enhance their bioavailability. If sun exposure is insufficient or inefficient, vitamin D supplements are required to maintain an optimal vitamin D status. There are no reliable sources of vitamin B12 in plant foods, for which a B12 supplementation is mandatory for all vegans.

Due to the rapid increase in popularity of vegan diets, healthcare providers must be aware of the characteristics of a complete vegan diet in order to advise their patients correctly. Vegan diets restricting energy intake, excluding one or more food groups, not paying attention to critical nutrients or to vitamin D status, and not supplementing vitamin B12 cannot be considered well-balanced, and may have dangerous health consequences.

This paper summarizes the recommendations made by the Scientific Society for Vegetarian Nutrition (SSNV), concerning vegan diets during these delicate phases of life. Since there are not enough studies to give evidence-based recommendations, the evidence level of such statements is to be considered as expert opinion.

Not following these recommendations can put these vulnerable subjects at clear risk for nutritional deficiencies.

 And added the following new references

97-Dagnelie, P.C. van Dusseldorp, M.; van Staveren, W.A.; Hautvast, J.G. Effects of macrobiotic diets on linear growth in infants and children until 10 years of age. Eur J Clin Nutr 1994, 48, S103–11 1, discussion S111-2.

98-Van Staveren, W.A. Dagnelie, P.C. Food consumption, growth, and development of Dutch children fed on alternative diets. Am J Clin Nutr 1988, 48, S819–21, doi: 10.1093/ajcn/48.3.819.

99-Van Dusseldorp, M.; Arts I.C.; Bergsma, J.S.; De Jong, N.; Dagnelie, P.C.; Van Staveren, W.A. Catch-up growth in children fed a macrobiotic diet in early childhood. J Nutr 1996, 126, 2977–83.

100-Crawford, J.R. Say, D. Vitamin B12 deficiency presenting as acute ataxia. BMJ Case Rep 2013, 26, 2013, doi: 10.1136/bcr-2013-008840.

101-Cundiff, D.K. Harris, W. Case report of 5 siblings: malnutrition? Rickets? DiGeorge syndrome? Developmental delay? Nutr J 2006, 5, 1.

102-Amoroso, S.; Scarpa, M.G.; Poropat, F.; Giorgi, R.; Murru, F.M.; Barbi, E. Acute small bowel obstruction in a child with a strict raw vegan diet. Arch Dis Child 2018, doi: 10.1136/archdischild-2018-314910.

  Round  2

Reviewer 1 Report

The paper has been well revised.

Reviewer 2 Report

all comments have been sufficiently adressed.